# Identification of Novel Hub Genes Associated with Psoriasis Using Integrated Bioinformatics Analysis

**DOI:** 10.3390/ijms232315286

**Published:** 2022-12-04

**Authors:** Qi Yue, Zhaoxiang Li, Qi Zhang, Quanxin Jin, Xinyuan Zhang, Guihua Jin

**Affiliations:** Department of Immunology and Pathogenic Biology, Yanbian University Medical College, Yanji 133002, China

**Keywords:** psoriasis, bioinformatics, molecular markers, immune microenvironment, potential drugs

## Abstract

Psoriasis is a chronic, prolonged, and recurrent inflammatory skin disease and the current therapeutics can only alleviate the symptoms rather than cure it completely. Therefore, we aimed to identify the molecular signatures and specific biomarkers of psoriasis to provide novel clues for psoriasis and targeted therapy. In the present study, the Gene Expression Omnibus (GEO) database was used to retrieve three microarray datasets (GSE166388, GSE50790 and GSE42632) and to explore the differentially expressed genes (DEGs) in psoriasis using the Affy package in R software. The gene ontology (GO) and Kyoto Encyclopedia of Gene and Genome (KEGG) pathway enrichment were utilized to determine the common DEGs and their capabilities. The STRING database was used to develop DEG-encoded proteins and a protein–protein interaction network (PPI) and the Cytohubba plugin to classify hub genes. Using the NetworkAnalyst platform, we detected transcription factors (TFs), microRNAs and drug candidates interacting with hub genes. In addition, the expression levels of hub genes in HaCaT cells were detected by western blot. We screened the up- and downregulated DEGs from the transcriptome microarrays of corresponding psoriasis patients. Functional enrichment of DEGs in psoriasis was mainly associated with positive regulation of leukocyte cell–cell adhesion and T cell activation, cytokine binding, cytokine activity and the Wnt signaling pathway. Through further data processing, we obtained 57 intersecting genes in the three datasets and probed them in STRING to determine the interaction of their expressed proteins and we obtained the critical 10 hub genes in the Cytohubba plugin, including *TOP2A, CDKN3, MCM10, PBK, HMMR, CEP55, ASPM, KIAA0101, ESC02*, and *IL-1β*. Using these hub genes as targets, we obtained 35 TFs and 213 miRNAs that may regulate these genes and 33 potential therapeutic agents for psoriasis. Furthermore, the expression levels of TOP2A, MCM10, PBK, ASPM, KIAA0101 and IL-1β were observably increased in HaCaT cells. In conclusion, we identified potential biomarkers, risk factors and drugs for psoriasis.

## 1. Introduction

Psoriasis is an immune-mediated, genetic disease characterized by the formation of scaly, indurated, erythematous plaques. The three main histologic characteristics of psoriasis are epidermal hyperplasia, dilated, conspicuous blood vessels in the dermis and an inflammatory leucocyte infiltrate, primarily into the dermis [1]. The underlying pathomechanisms involve a complex interaction between the innate and adaptive immune systems. T cells interact with dendritic cells, macrophages and keratinocytes, which can be mediated by their secreted cytokines [2]. The dermatologic manifestations of psoriasis are varied; psoriasis vulgaris is also called plaque−type psoriasis and is the most prevalent type [3]. The disease is often associated with psoriatic arthritis, metabolic syndrome, cardiovascular problems, diabetes mellitus and other comorbidities. Psoriasis patients have a higher risk for chronic inflammatory bowel disease and chronic kidney disorders. Moreover, the prevalence of depression, anxiety and suicidality are increased [4]. Taken together, different factors contribute to the development of psoriasis causing adverse effects on patients’ quality of life and disease burden. Therefore, exploring the possible mechanisms of psoriasis is of great importance.

Microarray and high-throughput sequencing have been considered reliable techniques to quickly detect differentially expressed genes (DEGs),these techniques are able to make various slice data, which are produced and stored in public databases. Consequently, many valuable clues could be explored for new research on the basis of these data [5]. Recently, numerous microarray gene profile studies on psoriasis have been carried out. The gene expression patterns in psoriasis have been revealed and explained with microarray data [6]. The integrated bioinformatics analysis will be more trustworthy and offer useful new molecular targets to support the creation of precise diagnoses and cutting-edge treatment plans.

The purpose of this study is to find crucial genes related to the pathogenesis of psoriasis. We analyzed three gene expression datasets (GSE166388, GSE50790, and GSE42632) downloaded from the Gene Expression Omnibus (GEO) database, dissecting the gene ontology (GO) and Kyoto Encyclopedia of Gene and Genome (KEGG) pathway enrichment utilized to determine the common DEGs and their capabilities in psoriasis. What’s more, the PPI network was developed for gene modules and to recognize center genes by utilizing the STRING database and Cytoscape software. Eventually, we distinguished 10 significant center genes and we further examined the miRNAs, potential drugs and corresponding transcription factors of these genes. The genes central to psoriasis identified here are expected to provide new insights into the biological mechanisms of this disease.

## 2. Results

### 2.1. Information of Expression Profiling Data

The gene expression profiles of the three datasets (GSE166388, GSE50790, and GSE42632) were recovered from those present in the GEO dataset. Among them, GSE166388 contained healthy human skin tissue (H) and lesion skin tissue of plaque psoriasis (PP), GSE50790 included non-lesional skin tissue of plaque psoriasis (PN) and PP and GSE42632 contained dermal MSCs derived from psoriasis patients and normal controls. As per the rules of *p* < 0.05 and |logFC| > 1.0, 1294 DEGs altogether were obtained from GSE166388, containing 608 upregulated genes and 686 downregulated genes. The 1155 DEGs were obtained when the gene chip GSE50790 was separated. Among them, 609 genes were upregulated and 546 genes were downregulated. In the GSE42632 dataset, 1487 DEGs were obtained, 908 of which were upregulated and 579 of which were downregulated. The upregulated DEGs were checkered red and those downregulated blue in volcano plots (Figure 1a). The top 50 significant upregulated genes and top 50 significant downregulated genes were represented by heatmaps of DEG expression in Figure 1b. To make our understanding of the data more in-depth, principal component analysis (PCA) was carried out. The DEGs of psoriasis and normal tissues were relatively well-separated in the Figure 1c score plot PCA. These DEGs may be associated with pathogenic cycles or clinical status. In this way, the identification of common DEGs might give important data to grasping psoriasis.

### 2.2. Analysis of the Functional Characteristics of Common DEGs

To investigate the biological processes and pathways involved with the common DEGs, GO and KEGG Pathway enrichment analyses were carried out. From Figure 2, we obtained some biological functions and signaling pathways associated with psoriasis. Biological processes, cellular components and molecular functions are included in the findings of the GO analysis. Within these parent categories, enriched categories were found. The most highly enriched biological processes were skeletal system morphogenesis, embryonic organ development, embryonic organ morphogenesis, translational termination, mitochondrial translational termination, mitochondrial translational elongation, response to lipopolysaccharide, positive regulation of leukocyte cell–cell adhesion and T cell activation (Figure 2). The most highly enriched cellular components were chromosomal region, condensed chromosome, collagen-containing extracellular matrix, mitochondrial protein complex, mitochondrial ribosome, organellar ribosome, an intrinsic component of synaptic membrane, glutamatergic synapse and external side of the plasma membrane (Figure 2). Single-stranded DNA binding, DNA helicase activity, extracellular matrix structural constituent, threonine−type peptidases, threonine−type endopeptidases, structural constituent of ribosome, chemokine activity, cytokine binding and cytokine activity were the most highly enriched molecular functions (Figure 2). Furthermore, KEGG pathway enrichment analysis showed focal adhesion, the Wnt signaling pathway, vascular smooth muscle contraction, non–alcoholic fatty liver, Alzheimer’s disease, ribosome, PPRP signaling pathway, viral protein with cytokine and cytokine receptor and cytokine–cytokine receptor interaction (Figure 2). In conclusion, these functional enrichment results may have some implication for the evolution of psoriasis. GO and KEGG analysis specifics for the three datasets are presented in Appendix A.

### 2.3. PPI Network Construction and Hub Genes Identification

To elucidate the correlation of differentially expressed genes in the three groups of psoriasis gene expression profiles, we obtained the intersection of the Venn diagram and obtained 57 common DEGs (Figure 3a and Appendix A). A construction of the PPI network was built from the 57 genes that may have prognostic value using the STRING network analysis tool and core genes were analyzed using the CytoHubba plugin in Cytoscape software. The PPI network was comprised of 54 nodes and 322 edges. The subnetwork with the most hubs and edges was displayed in Figure 3b. In light of the STRING dataset, Cytoscape was used to perform and display the PPI analysis of DEGs. We distinguished 10 focal genes in the PPI organization, including *TOP2A, CDKN3, MCM10, PBK, HMMR, CEP55, ASPM, KIAA0101, ESC02* and *IL-1β*, as potential center genes in light of the hub degree scores created through Cytoscape. These 10 hub genes were presented in Figure 3c and Table 1. These genes co-appeared between healthy people and psoriasis, psoriasis lesions and non-lesions, psoriasis and healthy mesenchymal cells and showed a special group with interaction, implying these central genes may have important roles in the pathogenesis, clinical treatment and metabolic processes of psoriasis.

### 2.4. The Candidate miRNA–Hub Gene Regulatory Network

To gain insight into the relationship between miRNAs and psoriasis-focused target genes during transcriptional repression or abrogation of protein translation, we developed essentially unique miRNAs and gene regulatory networks utilizing Cytoscape software. The targeted miRNAs were anticipated in view of NetworkAnalyst datasets. Figure 4 shows the hub genes and the associated regulatory miRNA molecules. The top five miRNAs were described in Table 2. From Table 2, *mir-34a-5p, mir-129-2-3p, mir-103a-3p* and *mir-124-3p* were found to be more associated with psoriasis. According to research, miR-34a-5p downregulation increases osteoarthritis cell proliferation while decreasing apoptosis and autophagy [7]. *MiR-129-2-3p* is engaged in the malfunctioning of diabetic neutrophils. Neutrophil retention kinetics and chronic inflammation may be triggered by *miR-129-2-3p* regulated genes such as Casp6 and Ccr2 [8]. Thus, these candidate miRNAs can provide a strong basis for understanding the molecular mechanisms of psoriasis and reveal a promising series of targets of psoriasis.

### 2.5. Immune Cell Infiltration Results

Psoriasis is a disease in which multiple immune cells are involved. We used the CIBERSORT algorithm to evaluate the immune infiltration of 22 inflammatory cell subsets in psoriatic-lesioned skin biopsies, healthy skin and non-lesioned skin biopsies in an attempt to investigate the role of immune cell infiltration in the pathobiology of psoriasis patients. The violin graph demonstrated (Figure 5a) that compared with the normal control sample, there were more CD8^+^ T cells, activated memory CD4^+^ T cells, follicular helper T cells, M2 macrophages, resting dendritic cells, activated dendritic cells and neutrophils in the psoriasis lesion tissue, but fewer memory B cells, plasma B cells, resting memory CD4^+^ T cells, regulatory T cells (Tregs), resting NK cells, activated NK cells, Monocyte, M0 macrophages, M1 macrophages and activated mast cells and eosinophils. Active specific inflammatory cells have negative implications for the evolution of the psoriasis pathogenesis process. There was not a prominent statistical difference in infiltrating abundance of immune cells between the lesioned skin and non-lesioned skin in patients (Figure 5b). The study of immune cell infiltration may provide new treatments for psoriasis and facilitate a more detailed and in-depth understanding of its pathogenesis.

### 2.6. Analysis of the Transcriptional Factor Signaling Pathways for a Selected Group of Genes

Psoriasis is a polygenic illness whose pathogenesis corresponds with transcription factors [9]. Hence, identifying the common STFs would help understand the possible mechanism linked to psoriasis. From the genes we identified, a gene–TF regulatory network was constructed including 129 interaction pairs among the selected genes and 35 TFs (Figure 6). *ESCO2, ASPM, HMMR, CDKN3* and *TOP2A* were found to be regulated by eight TFs, while *CEP55* was found to be regulated by seven TFs, *PBK* and *MCM10* by six TFs and IL1B by three TFs. In addition, various TFs were found to regulate more than one hub gene and fifteen TFs were identified with a connectivity degree ≥ 2 in the gene–TF regulatory network, which means that these TFs have close interactions with these hub DEGs. For example, nuclear factor I-C (NFIC) was predicted to regulate *MCM10, TOP2A, ESCO2, CDKN3* and *CEP55*; nuclear transcription factor Y, subunit A (NFYA) was found to regulate *TOP2A, ESCO2, CDKN3, IL-1β* and *HMMR.* Identifying the expression of key transcription factors is critical for directing psoriasis gene expression and participating in its physiopathological processes.

### 2.7. Drug–Hub Gene Interaction

The etiology of psoriasis is still unclear, there are no effective clinical treatments and drugs. To explore the interrelationship between the central genetic aspects of psoriasis and the available drugs, the hub gene–drug interaction network was constructed with NetworkAnalyst (http://www.networkanalyst.ca/faces/home.xhtml (accessed on 18 June 2022)) to visualize. Utilizing the 10 center genes to investigate the drug–gene cooperations, 33 medications for conceivably treating psoriasis were chosen and gathered (Figure 7 and Table 3). *TOP2A* was screened for a wide range of relationships with multiple drugs, which may provide potential targets for the treatment and prognosis of psoriasis.

### 2.8. Expression of Hub Genes in HaCaT Cells

To further explore the expression of differences of hub genes in psoriasis, HaCaT cells were treated with TNF-α to establish a psoriasis-like cell model. As the results showed, compared with the control group, the expression levels of TOP2A, MCM10, PBK, ASPM, KIAA0101 and IL-1β were significantly increased after TNF-α treatment (Figure 8).

## 3. Discussion

Psoriasis is a chronic, non-infectious disease that affects people of all ages with no predilection for sex. It is connected to a number of comorbidities and can affect the skin, nails, and joints [10]. Gene microarray technology can uncover a huge number of hereditary changes in illness progression, which might give expected potential focus to sicknesses. Three GEO datasets—GSE166388, GSE50790 and GSE42632 were used for this study’s DEG screening. The DEGs were then subjected to GO and KEGG pathway analysis using DAVID. According to the KEGG pathway results, the DEGs were primarily linked to focal adhesion, the Wnt signaling pathway, vascular smooth muscle contraction, Alzheimer’s disease, non-alcoholic fatty liver disease, ribosome, the PPAR signaling pathway, viral protein with cytokine and cytokine receptor and cytokine–cytokine receptor interaction. These findings also provide important hints for researching molecular connections in the development of psoriasis. Indeed, the Wnt signaling pathway did play a significant role in embryonic development and homeostatic self-renewal in adult tissues. Additionally, depending on the situation, it can also have pro- or anti-inflammatory effects and is controlled by a variety of mechanisms [11]. The principal controlling factor in the activation of the Wnt pathway is β-catenin. The Wnt/β-catenin pathway has an effect on both pro-inflammatory and anti-inflammatory responses, which are governed in various ways based on the conditions. In response to various stimuli, Wnt/catenin also variably affects NF-κB-mediated subsets of gene expression and biochemical characteristics (such as inflammation, cell proliferation and death). Disentangling the precise function of Wnt/catenin signaling in inflammation in the context of cell/tissue and physiology/pathology specifics will be significant [12]. Additionally, it had been shown in our earlier studies found that dihydroartemisinin (DHA) reduces imiquimod-induced psoriasis-like skin inflammation in mice and its potential mechanism is possibly to inhibit keratinocytes’ excessive cell division and the cytokines they secreted via the MAPK/NF-κB signaling pathway. The family of nuclear hormone receptors known as peroxisome-proliferator-activated receptors (PPARs) includes the PPARG, PPARD and PPARγ. The expression of PPARG and PPARD genes in the human epidermis was modulated by keratinocyte-derived IL-1 according to research [13]. PPARγ plays a significant useful role in the guideline of skin boundary porousness as an inhibitor of keratinocyte cell multiplication and a promoter of terminal separation of the epidermis. To the extent, psoriasis is an inflammatory skin sickness portrayed by epidermal hyperproliferation and abnormal keratinocyte differentiation, proteins engaged with PPARγ flagging can be viewed as possible focuses for therapy [14].

Gathering and coordinating each one of the genes’ encoding proteins in the genome by developing a PPI network has demonstrated value in the examination of numerous illnesses. Utilizing CytoHuba in Cytoscape 3.9.1 to break down the PPI results, 10 center point genes were acquired. Moreover, the western blot analysis suggested that TOP2A, MCM10, PBK, ASPM, KIAA0101 and IL-1β expression were markedly increased in HaCaT cells (Figure 8). The results indicated that hub genes were potential candidates for use as biomarkers of psoriasis in humans. Nuclear enzyme *TOP2A* (topoisomerase II) regulated DNA topological shape and cell cycle progression [15]. In contrast to normal tissues, psoriasis tissues showed low expression of *TOP2A* [16]. The cell cycle and proliferation are both greatly aided by cyclin-dependent kinase inhibitor 3 (*CDKN3*). When *CDKN3* binds to cyclin proteins, the CDK1 and CDK2 proteins are dephosphorylated, which prevents the cell cycle from continuing [17]. Numerous studies have been conducted on the dynamic expression of CDKN3 and its oncogenic involvement in various types of illnesses. *CDKN3* regulates the cell cycle and chemo-resistance in esophageal cancer, which promotes cancer progression [18]. By controlling the cell cycle and DNA replication signals, cyclin-dependent kinase inhibitor 3 (*CDKN3*) is a key player in the development of prostate cancer [19]. *MCM10* is unique to eukaryotes and is an important gene product that, when mutated, disrupts the entry into S-phase. *MCM10* is generally considered to act as an initiation factor. It is only required at the final step of initiation and is thought to activate the CMG helicase or help separate the two CMGs to achieve bidirectional replication forks. Other studies have shown that *MCM10* may also have function in replication forks [20,21]. Thus, according to research, *MCM10* inhibiting molecules can be employed to target breast cancer CSCs as well as tumors [22]. T-lymphokine-initiated executioner cell-initiated protein kinase (TOPK), otherwise called PDZ Binding Kinase (PBK), is a serine–threonine kinase of the mitogen-enacted protein kinase (MAPKK) family and an oncogenic protein that directs cell endurance, multiplication, development, apoptosis and irritation. TOPK expression and activation levels directly regulate the cell cycle. Indeed, TOPK could be a significant milestone in this development. PRC1, which is crucial for the development of the spindle and its equatorial plan, is activated by TOPK, which has an impact on the process of cell mitosis and encourages cell division. Prior research suggested that TOPK could bind to PRC1 via its C-terminal glutamate aspartic acid repetitive sequence to encourage PRC phosphorylation at the T481 site. This, in turn, increased the level of phosphorylation of CDK1/cyclin B1 to PRC1, which ultimately promoted cell cycle division [23,24]. Research has revealed that in mouse epidermis and human epidermal keratinocytes, a lack of autophagy reduces the expression of PBK, a regulator of the cell cycle (HEKs). Collectively, autophagy promotes epidermal growth, in part via controlling the expression of PBK [25].

Furthermore, a significant part of the development of illnesses was caused by the complex interactions between TFs and other hub genes. NFIC, NFYA, POU2F2, FOXA1 and SRF were discovered to be significant in psoriasis in our study after a gene-transcription factor regulatory network and several relevant transcription factors were evaluated. Finally, 33 medicines that may be effective in treating psoriasis were discovered. In Table 3, 10 hub genes, including *TOP2A*, were identified as possible pharmacological targets. The majority of medications are *TOP2A* inhibitors and are successful in treating the majority of inflammatory diseases. To ascertain whether these medications are effective in treating psoriasis, additional research and clinical trials are required. Nevertheless, this research may offer helpful insights into personalized and targeted psoriasis treatment as well as potential novel applications for traditional medications.

## 4. Materials and Methods

### 4.1. Differentially Expressed Gene (DEG) Selection

Public genome datasets include the GEO database (http://www.ncbi.nlm.nih.gov/geo (accessed on 6 June 2022)). In this study, three gene expression profiles, GSE50790 and GSE42632 were downloaded from the GEO. The explanation document inside the platform’s goal is to pair the probes with the relevant genes. GSE166388 contains transcriptomic information of four cases of PP and four cases of H. GSE50790 contains transcriptomic information of four cases of PP and four cases of PN. GSE42632 includes dermal MSCs derived from six psoriasis patients and six normal controls. Table 4 summarizes the information of the three selected datasets, GSE number, platform, sample, organization type, etc.

### 4.2. Data Preprocessing and Screening of Differentially Expressed Genes

We utilized the ComBat function of the sva package [29] of the R language to remove the collective effect between datasets to generate the common gene expression matrix and eliminate variability among datasets. Additionally, the Affy package [30] was used to perform normalization, background correction and expression calculation on the collected data. The probes were then annotated using a chip platform annotation file and matrix data. If different probes had the same average value and were linked to the same mRNA, that level of gene expression would be taken into account.

After data preprocessing, the DEGs between experimental and control groups for each data were detected by the limma function in the R program. The DEGs were chosen based on the criteria of |logFC| > 1.0 and a corrected *p*-value of 0.05. Finally, the volcano plot and PCA plot were depicted by R to demonstrate the differentiation between the groups of each dataset.

### 4.3. Gene Ontology and Pathway Enrichment Analysis

To further understand the corresponding differential genes involved in the regulation of those biological processes and signal transduction pathways, the corresponding genes were subjected to GO and KEGG analysis [5]. Cellular component (CC), molecular function (MF) and biological process (BP) are the three basic domains that GO uses to characterize the characteristics of the chosen genes [31]. KEGG (Kyoto Encyclopedia of Genes and Genomes) was originally designed as an integrated database resource for interpreting fully sequenced genomes through KEGG pathway maps, which are procedures for mapping to manually generate route maps using genes in the genome [32]. In this investigation, significantly upregulated and downregulated DEGs combined with psoriasis microarray data were analyzed by the R language (cluster profile package [version 3.14.3], Org.hs.eg. DB package [version 3.10.0] (for ID conversion)). The detailed data used the ggplot2 package to demonstrate.

### 4.4. Protein–Protein Interaction (PPI) and Establishment and Identification of Hub Genes

We extracted the intersection of the differential genes of these three groups of DGEs and uploaded the intersection gene list to the Multiple Protein Network Tool in the STRING dataset (https://string-db.org/ (accessed on 10 June 2022)), setting the composite score > 0.4 as the cut-off in order to further understand the interaction between the corresponding differential genes and screen the significant molecules. The TSV documents were downloaded and submitted to Cytoscape software, an open-source software stage for envisioning complex organizations, to visualize PPI networks. By examining the protein–protein interaction networks’ topology, each of the 12 centrality numerical estimations in cytoHubba, a java module in Cytoscape, was used to screen for focal genes.

### 4.5. Evaluation of Immune Cell Infiltration

Psoriasis is a long-term immune-related disease and the activity of related immune cells plays an important role in its pathogenesis. To understand the immune infiltration of our tissue samples accordingly, we took the corresponding deconvolution algorithm to estimate the presence of different immune cell subsets in the tissue.

The associated cellular immune infiltration algorithm was passed through CIBERSORT Methods. The marker information of 22 immune cells was extracted to calculate the infiltration of immune cells in each dataset. The corresponding molecular markers and algorithm information were from a Nature article (Robust enumeration of cell subsets from tissue expression profiles) [33,34].

### 4.6. The Network of miRNAs Associated with Hub Genes

Using NetworkAnalyst 3.0 (https://www.networkanalyst.ca/ (accessed on 18 June 2022)), an online visualization tool that aids in discovering miRNA–gene connections in gene regulatory networks, the hub genes were mapped to the matching miRNAs. MiRNAs with a degree cutoff value of 1.0 were discovered for each hub gene. Finally, Cytoscape software was used to map these hub genes and miRNAs.

### 4.7. TF Regulatory Network Construction

To investigate TF–gene interactions for the input genes and evaluate the impact of the TF on the expression and functional pathways of the hub gene, NetworkAnalyst (http://www.networkanalyst.ca/faces/home.xhtml (accessed on 18 June 2022)) was also utilized. In this study, the TFs of the hub genes were predicted from this database, and the Cytoscape program was used to build and show a transcriptional regulatory network.

### 4.8. Drug–Hub Gene Interaction

The medication quality communication dataset (DGIdb) is an open public data stage for known and potential medication quality associations [35]. Drug databases were limited to the Food and Drug Administration (FDA) and DrugBank. To screen potential drugs targeting psoriasis, the corresponding hub genes were mapped into the DrugBank network. The drug–gene interaction network was visualized using NetworkAnalyst (http://www.networkanalyst.ca/faces/home.xhtml (accessed on 18 June 2022)). To determine whether related clinical trials are reporting these potential drugs for psoriasis, these identified drugs were input into the ClinicalTrials.gov registry (https://clinicaltrials.gov/ (accessed on 20 June 2022)) and pharmsnap (https://pharmsnap.zhihuiya.com/ (accessed on 20 June 2022)), which are two highly used, widely trusted sources of new drugs and drug trials worldwide.

### 4.9. Cell Line and Treatment

HaCaT cells were purchased from FuDan IBS Cell Center (FDCC-HPN096, FDCC, Shanghai, China). HaCaT cells were stimulated with 50 ng/mL recombinant human TNF-α (50 ng/mL; Peprotech, NJ, USA) for 24 h.

### 4.10. Western Blot Analysis

After indicated treatment, cells were harvested and washed twice with PBS. Then they were lysed in RIPA buffer (Solarbio, Beijing, China) containing complete protease and phosphatase inhibitor (Solarbio, Beijing, China). The level of protein concentrations in cells was measured with a bicinchoninic acid protein assay kit (Solarbio, Beijing, China). The proteins were isolated by sodium dodecyl sulfate-polyacrylamide gel electrophoresis (SDS-PAGE) and transferred to polyvinylidene fluoride (PVDF) membranes by a wet electrophoretic transfer method. The membrane was blocked with TBST containing 5% non-fat milk for 2 h at room temperature followed by incubation overnight at 4 °C with primary antibodies. After overnight incubation, the blots were washed three times with TBST and incubated with secondary antibodies for 1 h at room temperature. Finally, the membranes were scanned using a Bio-Rad Gel imaging system (Bio-Rad, Berkeley, CA, USA) after visualization treatment with the ECL reagent. The primary antibodies, rabbit anti-PBK, rabbit anti-KIAA0101, rabbit anti-TOP2A, rabbit anti-IL-1β, and rabbit anti-β-actin, were purchased from Cell Signaling Technology (Boston, MA, USA). The rabbit anti-ASPM and rabbit anti-MCM10 were obtained from Proteintech (Rosemont, Chicago, IL, USA). Quantitative assessment of the and intensity was performed using Image Lab statistical software.

### 4.11. Statistical Analysis

All data were analyzed with GraphPad Prism 7.0 (GraphPad Software, La Jolla, CA, USA) and exhibited as the means ± SEM. The significance of differences was determined using the Formulation *t*-test; *p* < 0.05 was considered statistically significant.

## 5. Conclusions

Through comprehensive bioinformatic analysis, this study identified DEGs normal in psoriasis; that is, gathering in DNA replication, cell cycle, DCC-intervened pathways, and Netrin-1 flagging pathways. We also identified 10 central genes that may play an important role in psoriasis, including *TOP2A, CDKN3, MCM10, PBK, HMMR, CEP55, ASPM, KIAA0101, ESC02* and *IL-1β*. These 10 central genes may serve as new target markers for early detection, prognosis and targeted therapy in psoriasis. Also, potential miRNA and transcription factors were screened. In addition, a group of drugs was identified that could potentially be used to treat patients with psoriasis. This study provides a strong foundation for psoriasis research and requires in-depth experimental studies.

## Figures and Tables

**Figure 1 ijms-23-15286-f001:**
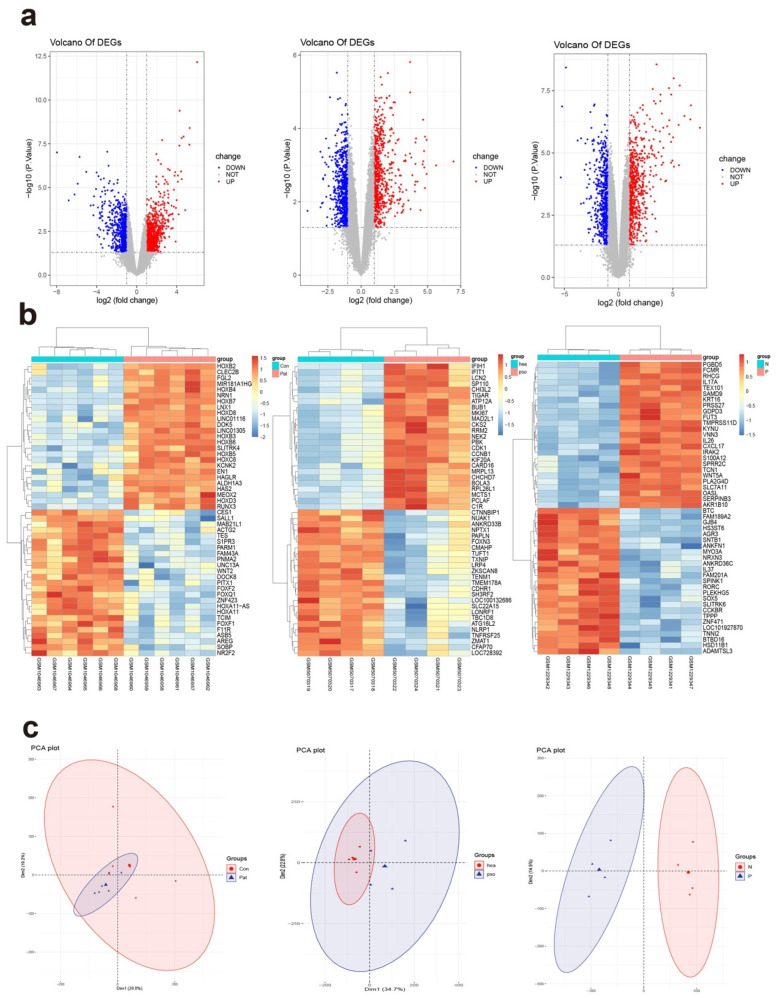
(**a**) Volcano plot representing DEGs among normal control group, PN and PP. The *X*-axis indicates the adjusted *p* (log-scaled) and the *Y*-axis shows the fold change (log-scaled). Each dot stands for a separate gene; red dots represent genes that have been upregulated, gray spots represent genes that have had little to no change in expression, and blue dots represent genes that have been downregulated based on adjusted *p* < 0.05 and |log2 FC| > 1. (**b**) The DEGs’ heat map. The sample numbers are located in the abscissa, while the gene names are in the ordinate. DEGs that are upregulated are indicated by red, while those that are downregulated by blue. (**c**) PCA among normal control group, PN and PN. The smaller the overlap of two circles, the more different of two groups.

**Figure 2 ijms-23-15286-f002:**
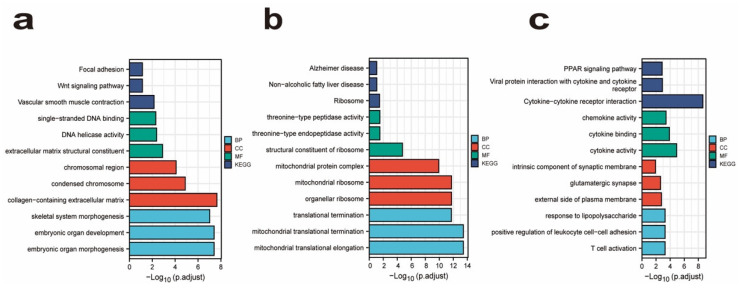
GO functional and KEGG pathway enrichment analysis of DEGs. GO functional analysis showing enrichment of DEGs in biological process (**a**), molecular function (**b**) and cellular component (**c**). KEGG pathway enrichment analysis of DEGs.

**Figure 3 ijms-23-15286-f003:**
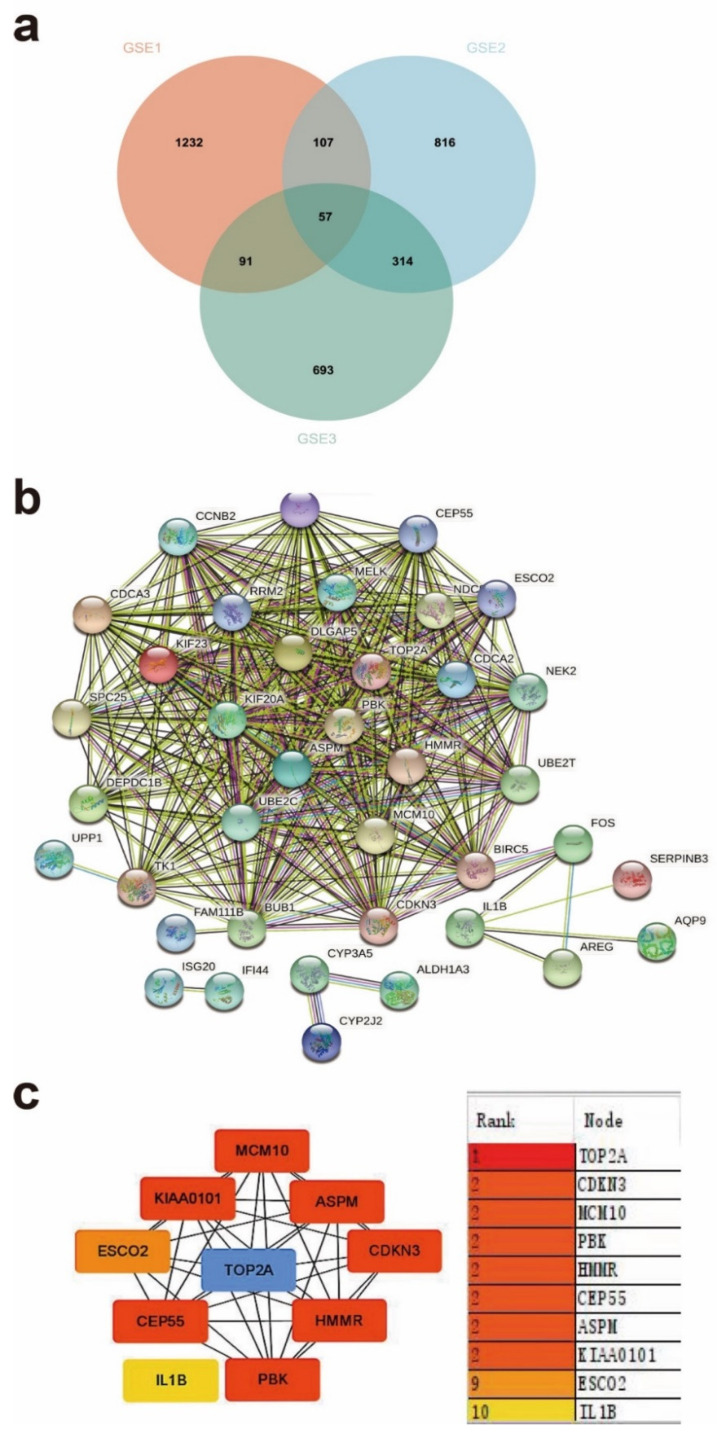
(**a**) Venn diagrams showing the overlaps of numbers of DEGs between three selected GEO datasets. (**b**) PPI network investigation. The interaction evidence degree between proteins is presented as the grayscale of the lines. (**c**) The top 10 hub genes were calculated by MCC in Cytoscape. Colors represent the importance of genes. The blue color stands for the highest degree, and the yellow color represents the lowest degree.

**Figure 4 ijms-23-15286-f004:**
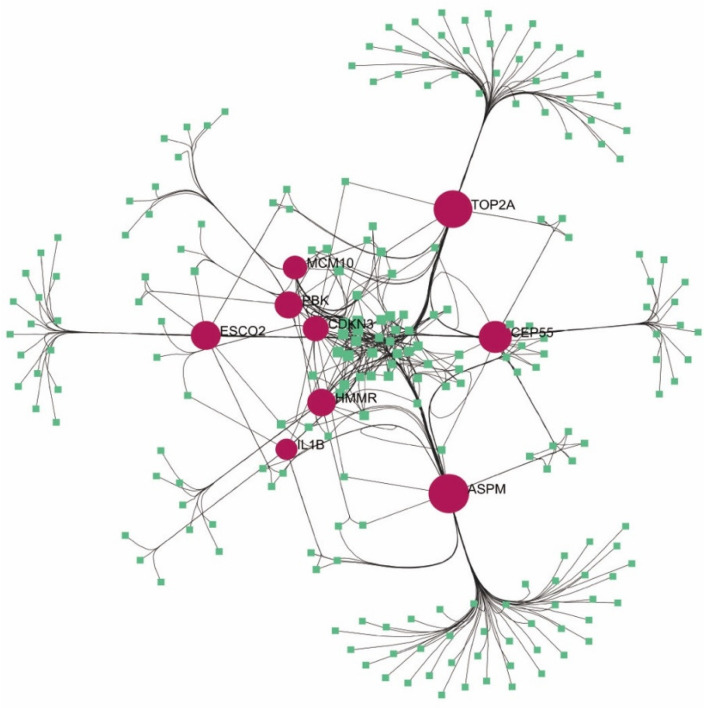
Integrated miRNA–DEG networks for the hub genes. The hub gene was shown in red circle, while the miRNA was in green node.

**Figure 5 ijms-23-15286-f005:**
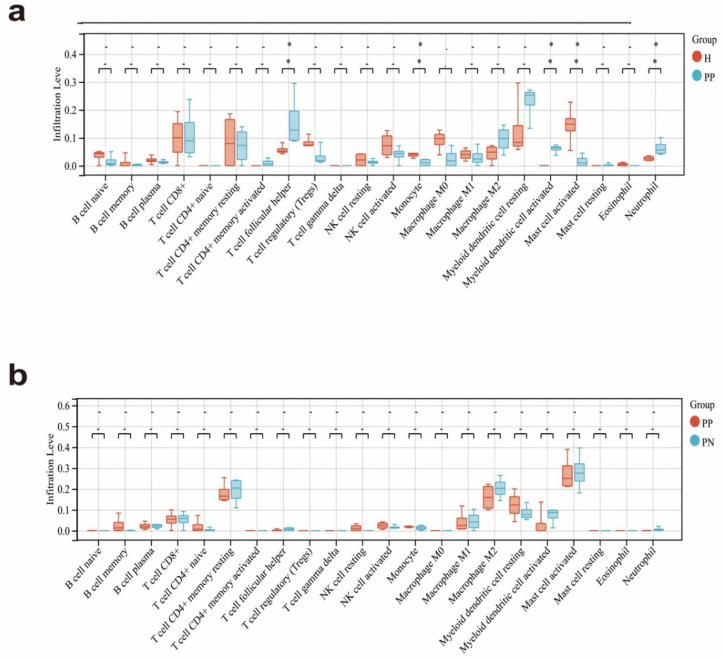
22 immunological cells were represented on a violin chart. (**a**) Data was from PP and normal control groups and (**b**) Data was from PP and PN. The black mark denoted a substantial distinction between the two sample groups. * *p* < 0.05.

**Figure 6 ijms-23-15286-f006:**
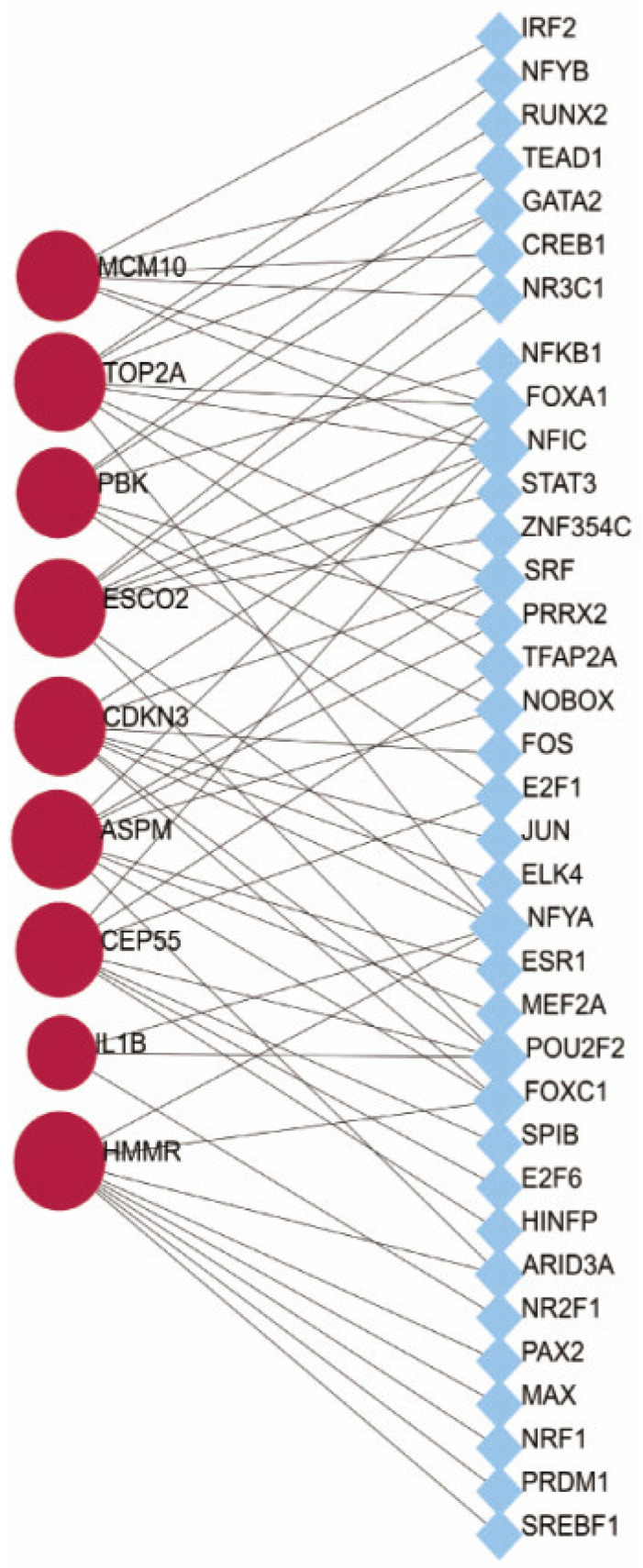
The hub gene–TF regulatory network. The red circle standed for the hub gene and the blue node standed for the transcription factor.

**Figure 7 ijms-23-15286-f007:**
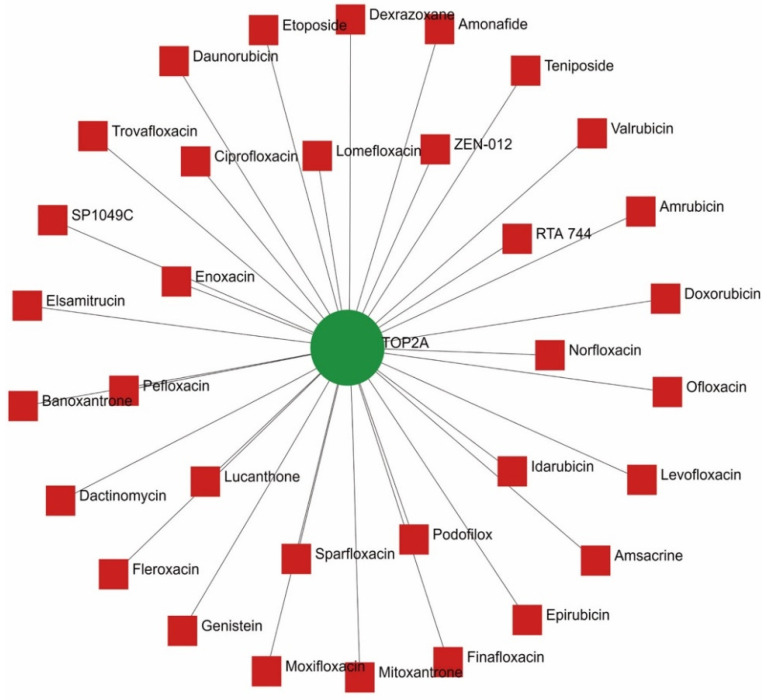
Drug–Hub gene interaction. The green circle represented the hub gene; the red squares represented the drugs.

**Figure 8 ijms-23-15286-f008:**
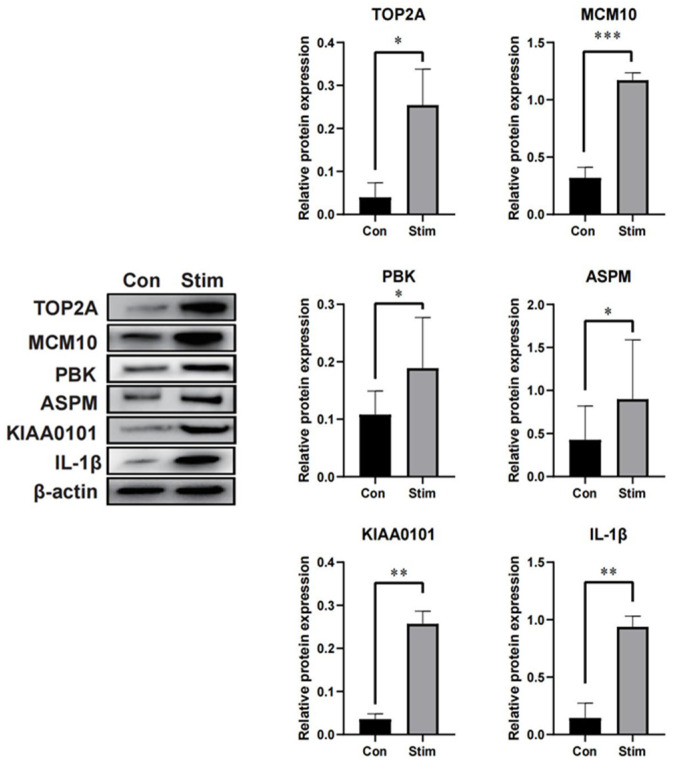
HaCaT cells were treated with recombinant human TNF-α for 24 h. The expression levels of TOP2A, MCM10, PBK, ASPM, KIAA0101 and IL-1β were detected by western blot. The data are represented as means ± SEM; *n* ≥ 3 in all experiments. * *p* < 0.05, ** *p* < 0.01, *** *p* < 0.001.

**Table 1 ijms-23-15286-t001:** Information on 10 hub genes.

Gene Symbol	Degrees	Full Name	Location	Highest Median Expression	Function	Ensembl Gene ID
*TOP2A*	18	Gene topoisomerase IIαaa	hg38 chr17:40388516-40417950	150.75 TPM in Cells-EBV-transformed lymphocytes	Not only can this gene encode DNA topoisomerases, but it can also control and alter the topological states of DNA during transcription.	ENSG00000131747.14
*CDKN3*	16	Cyclin-dependent kinase inhibitor-3	hg38 chr14:54396849-54420218	176.20 TPM in Testis	The *CDKN3* gene encodes a dual-specificity protein phosphatase that has a positive role in regulating cell proliferation.	ENSG00000100526.19
*MCM10*	16	Minichromosome maintenance protein 10	hg38 chr10:13161554-13211104	17.44 TPM in Cells-EBV-transformed lymphocytes	In order to encourage DNA replication and prevent replication stress, *MCM10* acts as a crucial scaffold.	ENSG00000065328.16
*PBK*	16	PDZ-binding kinase	hg38 chr8:27809620-27838095	81.37 TPM in Testis	The protein that controls how cells proliferate and how the cell cycle develops.	ENSG00000168078.9
*HMMR*	16	Hyaluronan mediated motility receptor	hg38 chr5:163460203-163491945	56.76 TPM in Cells-EBV-transformed lymphocytes	It is an intracellular, microtubule-associated, spindle assembly factor that localizes protein complexes to augment the activities of mitotic kinases and control dynein and kinesin motor activities.	ENSG00000072571.19
*CEP55*	16	Centrosomal protein 55 kDa	hg38 chr10:93496632-93529092	54.03 TPM in Cells-EBV-transformed lymphocytes	It is a crucial regulator of cytokinesis, the final stage of mitotic cell division.	ENSG00000138180.15
*ASPM*	16	Abnormal spindle-like microcephaly-associated	hg38 chr1:197084128-197146694	25.05 TPM in Cells-EBV-transformed lymphocytes	Abnormal spindle-like microcephaly-associated (*ASPM*) gene encodes a spindle protein that is commonly implicated in primary microcephaly.	ENSG00000066279.17
*KIAA0101*	16	Proliferating cell nuclear antigen binding factor	hg38 chr15:64364311-64387687	29.71 TPM in Cells-EBV-transformed lymphocytes	The protein regulates DNA synthesis and cell cycle progression.	ENSG00000166803.11
*ESCO2*	14	Establishment of cohesion 1 homolog 2	hg38 chr8:27771949-27812640	17.01 TPM in Cells-EBV-transformed lymphocytes	It is an evolutionarily conserved cohesion acetyltransferase that exerts essential functions in the establishment of sister chromatid cohesion.	ENSG00000171320.14
*IL-1β*	6	Interleukin 1 beta	hg38 chr2:112829751-112836903	24.53 TPM in Whole Blood	*IL-1β* are important proinflammatory cytokines, as it has powerful proinflammatory activities and can promote the secretion of a variety of proinflammatory mediators.	ENSG00000125538.11

**Table 2 ijms-23-15286-t002:** Top five candidate miRNAs targeting hub genes.

ID	mi-RNA	Degree
MIMAT0000255	hsa-mir-34a-5p	9
MIMAT0004605	hsa-mir-129-2-3p	9
MIMAT0000101	hsa-mir-103a-3p	8
MIMAT0000422	hsa-mir-124-3p	8
MIMAT0000251	hsa-mir-147a	8
MIMAT0000416	hsa-mir-1-3p	7
MIMAT0000104	hsa-mir-107	7
MIMAT0000445	hsa-mir-126-3p	7
MIMAT0000069	hsa-mir-16-5p	7
MIMAT0000266	hsa-mir-205-5p	7
MIMAT0000646	hsa-mir-155-5p	7
MIMAT0000686	hsa-mir-34c-5p	6
MIMAT0000222	hsa-mir-192-5p	6
MIMAT0000461	hsa-mir-195-5p	6
MIMAT0000418	hsa-mir-23b-3p	6
MIMAT0000259	hsa-mir-182-5p	6
MIMAT0003327	hsa-mir-449b-5p	6
MIMAT0000264	hsa-mir-203a-3p	6
MIMAT0000424	hsa-mir-128-3p	5
MIMAT0001541	hsa-mir-449a	5
MIMAT0000082	hsa-mir-26a-5p	5

**Table 3 ijms-23-15286-t003:** Candidate drugs targeting hub genes.

ID	Drug Trial Phase	Drug	Interaction Types	Indications
DB00218	Clinical phase iii	RTA-744	Inhibitor	Glioblastoma Meningeal carcinomatosis Meningitis
DB00276	Termination	ZEN-012	Inhibitor	-
DB00380	Approved	NORFLOXACI	Inhibitor	Gonorrhea Prostatitis Typhoid Urinary tract infection
DB00385	Approved	IDARUBICIN	-	Breast Cancer Acute lymphoblastic leukemia Acute myeloid leukemia
DB00444	Approved	PODOFILOX	Inhibitor	Condyloma acuminatum
DB00445	Approved	SPARFLOXACIN	-	Pneumonia Respiratory infections Bacterial infections
DB00467	Termination	LUCANTHONE	-	-
DB00487	Approved	PEFLOXACIN	Inhibitor	Infectious Diseases
DB00537	Approved	ENOXACIN	Inhibitor	Bacterial Infections
DB00685	Approved	CIPROFLOXACIN	-	Pseudomonas infection Staphylococcal infections Otitis externa Otitis media Plague
DB00694	Approved	LOMEFLOXACIN	Inhibitor	Conjunctivitis Bacterial infections Bronchial infections Surgical wound infection Urinary tract infection
DB00773	Approved	TENIPOSIDE	Inhibitor	Acute lymphocytic leukemia Astrocytoma Bladder cancer Glioma Lymphoma
DB00970	Approved	VALRUBICIN	Inhibitor	Carcinoma in situ Bladder cancer
DB00978	Approved	AMRUBICIN	Inhibitor	Non-small cell lung cancer Small cell lung cancer
DB00997	Clinical phase iii	DOXORUBICIN	Inhibitor	Liver Cancer Small cell lung cancer
DB01059	Approved	OFLOXACIN	Inhibitor	Conjunctivitis Corneal ulcers Otitis externa Otitis media Bacterial infections
DB01137	Approved	LEVOFLOXACIN	Inhibitor	Chronic bronchitis Cellulitis Corneal ulcers
DB01165	Approved	AMSACRINE	Inhibitor	Leukemia
DB01177	Approved	EPIRUBICIN	Inhibitor	Alveolar soft tissue sarcoma Colorectal cancer Esophageal cancer Melanoma Pancreatic Cancer
DB01179	Approved	FINAFLOXACIN	Inhibitor	Abdominal infection Pneumonia Blepharitis
DB01204	Approved	MITOXANTRONE	Inhibitor	Peripheral T-cell lymphoma Multiple sclerosis Small cell lung cancer Breast cancer
DB01208	Approved	MOXIFLOXACIN	-	Abdominal infection Pneumonia Blepharitis
DB01645	Clinical phase ii	GENISTEIN	Inhibitor	Lung Injury Pulmonary fibrosis Bladder Cancer Prostate cancer Non-small cell lung cancer
DB04576	Approved	FLEROXACIN	-	Bacterial Infections
DB04967	Approved	DACTINOMYCIN	-	Choriocarcinoma Hodgkin’s lymphoma Neuroblastoma Ewing Sarcoma
DB04975	Termination	BANOXANTRONE	Inhibitor	-
DB04978	Termination	ELSAMITRUCIN	Inhibitor	-
DB05022	-	SP1049C	-	-
DB05129	Approved	TROVAFLOXACIN	-	Bacterial Infections Reproductive tract infections
DB05920	Approved	DAUNORUBICIN	Inhibitor	Acute myelogenous leukemia Myelodysplastic syndrome Acute lymphocytic leukemia
DB06042	Approved	ETOPOSIDE	Inhibitor	Ovarian Cancer Uterine Cancer Bladder Cancer Leukemia Lung Cancer
DB06263	Approved	DEXRAZOXANE	-	Cytoprotective adjuvants Cardiotoxicity A
DB09047	Termination	AMONAFIDE	-	-

**Table 4 ijms-23-15286-t004:** Information for psoriatic GEO data.

Dataset	Platform	Samples	Last Update Date	Reference
GSE166388	GPL570	4 PP and 4 H	24 August 2021	Qiu et al., 2021 [26]
GSE50790	GPL570	4 PP and 4 PN	25 March 2019	Swindell et al., 2012 [27]
GSE42632	GPL13497	6 PP and 6 PN	9 January 2018	Niu and Zhang, 2016 [28]

PP, lesion skin tissue of plaque psoriasis; PN, non-lesional skin tissue of plaque psoriasis; H, healthy human skin tissue.

## Data Availability

The three microarray psoriasis data sets were obtained from the GEO database (https://www.ncbi.nlm.nih.gov/geo/ (accessed on 6 June 2022)) with accession numbers GSE166388, GSE50790 and GSE42632. Information of PPI was available at https://cn.string-db.org/cgi (accessed on 10 June 2022). The network of miRNAs, TF and drugs associated with hub genes were available at https://www.networkanalyst.ca/NetworkAnalyst/home.xhtml (accessed on 18 June 2022). Identified drugs were input into the ClinicalTrials.gov registry (https://clinicaltrials.gov/ (accessed on 20 June 2022)) and pharmsnap (https://pharmsnap.zhihuiya.com/ (accessed on 20 June 2022)).

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
