# Peer review of "Identification of Novel Hub Genes Associated with Psoriasis Using Integrated Bioinformatics Analysis"

_ijms, 2022, doi:10.3390/ijms232315286_

Round 1

Reviewer 1 Report

Integrational analysis of gene expression studies is prevalent scientific topic in the context of psoriasis. Over the years plenty of comprehensive publications encompassing analysis based on reasonably selected data was made up. Presented manuscript is based on (merely) 3 studies, with no explanation about the choice concept, what is the part of manuscript major deficiency - lack of clarification of most of purposes, experiments, actions, conclucions throughout all sections. Out of many ambiguities there are major like: lack of precise distinction between non-lesional Ps skin and skin of healthy person, misleading definition "healthy skin" impedes understanting of presented data; unclear figures' and tables' description, like "Table 2. (should be named Table 1): Details of 10 hub genes" v.147; Figure 2 separated for 3 boxes making it hard to analyze; GO terms presented without IDS; lack of genes overlaped in Venn diagrams (Fig.3), v.141; desription of function in Table 2 (should be Table 1) should be short and informative; no information about the source of material from 2.5 section "Immune Cell Infiltration Result", etc. Basic concept and used methods are proper and interesting but the manuscript should be rewritten paying attention on elucidation of assumptions, aims and significance of experiments. It sould also undertake proofreading. 

Author Response

Thank you for your perspective comments on our paper, which are most helpful and valuable for improving the manuscript. After reconsidering seriously, our replies to the comments are as following. In addition, major revisions are indicated in red letters in the revised manuscript.

Q1:

Lack of precise distinction between non-lesional Ps skin and skin of healthy person, misleading definition "healthy skin" impedes understanting of presented data.

Our responses: 

We have modified the corresponding parts of the manuscript to make the expression more precisely. Non-lesional Ps skin represents the non-lesional skin tissue of plaque psoriasis (PN). Healthy skin represents skin of a healthy person. Major revisions are indicated in red letters in the revised manuscript.

Result, v. 91-: GSE166388 contained lesion skin tissue of plaque psoriasis (PP) and healthy skin (H), GSE50790 included plaque psoriasis non-lesional skin tissue (PN) and lesion skin tissue of plaque psoriasis (PP).

Q2:

Unclear figures' and tables' description, like "Table 2. (should be named Table 1): Details of 10 hub genes" v.147.

Our responses: 

As the reviewer’s suggested, we have changed the order of the Tables in the manuscript.

Q3:

Figure 2 separated for 3 boxes making it hard to analyze; GO terms presented without IDS.

Our responses: 

We have described each of the 3 boxes in Figure 2 in detail, such as Supplementary Table1, Table2, and Table3. These Tables have been inserted into the manuscript as supplementary data, v.168-.

Q4:

Lack of genes overlaped in Venn diagrams (Fig.3), v.141; desription of function in Table 2 (should be Table 1) should be short and informative.

Our responses:

As well as the reviewer’s suggested, we provide details of the 57 genes that overlap in the Venn diagram (Figure 3); as in Supplementary Table 4, v.206-.

Q5:

No information about the source of material from 2.5 section "Immune Cell Infiltration Result", etc.

Our responses: 

According to the revision, the information about source of material from 2.5 section "Immune Cell Infiltration Result" was actually shown in section 4.5 "Evaluation of Immune Cell Infiltration" in the manuscript. Major revisions are indicated in red letters in the revised manuscript. 

Materials and Methods, v. 560-: The associated cellular immune infiltration algorithm was passed through CIBERSORT Methods. The marker information of 22 immune cells was extracted to calculate the infiltration of immune cells in each data set. The corresponding molecular markers and algorithm information were from a Nature article (Robust enumeration of cell subsets from tissue expression profiles).  

Reference

[1]. Newman AM, Liu CL, Green MR, et al. Robust enumeration of cell subsets from tissue expression profiles. Nat Methods. 2015;12(5):453-457. doi:10.1038/nmeth.3337.

Q6:

Basic concept and used methods are proper and interesting but the manuscript should be rewritten paying attention on elucidation of assumptions, aims and significance of experiments. It sould also undertake proofreading. 

Our responses: 

Thank you for your acknowledgment of this manuscript. We have a re-examination on our paper carefully. With some revises, we hope to make our work more accurate. Major revisions are indicated in red letters in revised manuscript. 

Abstract, v. 14-: Identification of hub genes and differentially expressed genes (DEGs), as well as analyses of bio-logical pathway enrichment, gene ontology (GO) terminology and the creation of pro-tein-protein interaction (PPI) networks, were all examples of bioinformatics methodologies and identification of hub genes.

Introduction, v. 38-: The three main histologic characteristics of psoriasis are epidermal hyperplasia; dilated, conspicuous blood vessels in the dermis; and an inflammatory leucocyte infiltrate, primarily into the dermis [1].

Introduction, v. 59-: Recently, numerous microarray gene profile studies on psoriasis have been carried out. The gene expression patterns in psoriasis have been revealed and explained with microarray data [6]. The integrated bioinformatics analysis will be more trustworthy and offer useful new molecular targets to support the creation of precise diagnoses and cutting-edge treatment plans.

Introduction, v. 77-: Eventually, we distinguished 10 significant center genes, and further examined the mi-RNAs, potential drugs, and corresponding transcription factors of these genes.

Result, v. 85-: The gene expression profiles of the three datasets (GSE166388, GSE50790, and GSE42632) were recovered from those present in the GEO data set.

Result, v. 100-: The top 50 significant up-regulated genes and top 50 significant down-regulated genes were represented by DEGs expression heatmaps in Fig. 1b. To make our understanding of the data more in-depth, principal component analysis (PCA) was carried out. 

Result, v. 130-: To investigate the biological processes and pathways involved with the common DEGs, GO and KEGG Pathway enrichment analyses were carried out.  

Result, v. 134-: Biological processes, cellular components, and molecular functions are included in the findings of the GO analysis. Within these parent categories, enriched categories were found.

Result, v. 157-: GO and KEGG analysis specifics for the three datasets were presented in Supplementary Tables 1, 2, and 3.

Result, v. 184-: We distinguished 10 focal genes in the PPI organization, including TOP2A, CDKN3, MCM10, PBK, Gee, CEP55, ASPM, KIAA0101, ESC02, and IL-1β, as potential center genes in light of the hub degree scores created through Cytoscape.

Result, v. 212-: To gain insight into the relationship between miRNAs and psoriasis-focused target genes during transcriptional repression or abrogation of protein translation, we developed essentially unique miRNAs and gene regulatory networks utilizing Cytoscape software. The objective miRNAs were anticipated in view of NetworkAnalyst data sets. Figure 4 shows the top 9 DEGs and the associated regulatory miRNA molecules.

Result, v. 246-: We used the CIBERSORT algorithm to evaluate the immune infiltration of 22 inflammatory cells subsets in psoriatic lesioned skin biopsies, healthy skin, and non-lesioned skin biopsies in an attempt to investigate the role of immune cell infiltration in the pathobiology of psoriasis patients.

Result, v. 303-: Utilizing the 10 center genes to investigate the drug-gene cooperations, 33 medications for conceivably treating psoriasis were gathered and chosen (Fig. 7 and Table 3).

Discussion, v. 321-: Gene microarray technology can uncover a huge number of hereditary changes in illness progression, which might give expected potential focus to sicknesses.

Discussion, v. 341-: The principal controlling factor in the activation of the Wnt pathway is β-catenin. The Wnt/β-catenin pathway has effect on both proinflammatory and antiinflammatory, which are governed in various ways based on the conditions. In response to various stimuli, Wnt/-catenin also variably affects NF-B-mediated subsets of gene expression and biochemical characteristics (such as inflammation, cell proliferation, and death). Disentangling the precise function of Wnt/-catenin signaling in inflammation in the context of cell/tissue and physiology/pathology specifics will be significant [12].

Discussion, v. 355-: Additionally, it had been shown in our earlier studies that dihydroartemisinin (DHA) reduces imiquimod-induced psoriasis-like skin inflammation in mice, and its potential mechanism maybe inhibit keratinocytes' excessive cell division and the cytokines they secreted via the MAPK/NF-B signaling pathway.

Discussion, v. 444-: Furthermore, a significant part of the development of illnesses were caused by the complex interactions between TFs and other hub genes. NFIC, NFYA, POU2F2, FOXA1, and SRF were discovered to be significant in psoriasis in our study after a gene-transcription factor regulatory network and several relevant transcription factors were evaluated. Finally, 33 medicines that may be effective in treating psoriasis were discovered. In Table 4, 10 hub genes, including TOP2A, were identified as possible pharmacological targets. The majority of medications are TOP2A inhibitors and are successful in treating the majority of inflammatory diseases. To ascertain whether these medications are effective in treating psoriasis, additional research and clinical trials are required. Nevertheless, this research may offer helpful insights into personalized and targeted psoriasis treatment as well as potential novel applications for traditional medications.

Materials and Methods, v. 481-: GSE166388 contains transcriptomic information of four cases of plaque psoriasis lesioned skin tissue (PP) and four cases of healthy human skin tissue (H). GSE50790 contains transcriptomic information of four cases of plaque psoriasis lesional skin tissue (PP) and four cases of plaque psoriasis non-lesional skin tissue (PN).

Materials and Methods, v. 497-: We utilized the ComBat function of the sva package [26] of R language to remove the collective effect between datasets to generate the common gene expression matrix and eliminate variability among datasets. Additionally, the Affy package [27] was used to perform normalization, background correction, and expression calculation on the collected data. The probes were then annotated using a chip platform annotation file and matrix data. If different probes had the same average value and were linked to the same mRNA, that level of gene expression would be taken into account

Reviewer 2 Report

The authors aim to identify hub genes related to the pathogenesis of psoriasis and evaluated them in publicly available databases. In the end, A total of 10 genes were selected as pivotal genes, and TOP2A was strongly associated with the survival of patients. Besides, the author also did some enrichment analysis and drug prediction for the treatment of psoriasis. Though the study looks quite similar to many other recent studies utilizing bioinformatics analysis to discuss the relationship between genes and disease prognosis, the manuscript itself is quite accomplished and delivers some useful information. I have several comments for the authors’ consideration to further improve the manuscript.

1. In the analysis of the differentially expressed genes (DEGs) analysis, the author used a threshold as |log2FC| ≥ 1.0, I suggest the author use a higher cutoff like 2 or 2.5 to show a higher uniqueness of these DEGs genes.

2. In the functional enrichment analysis, the author only used an online website called DAVID to do the analysis. I suggest the author could also a try gene set enrichment analysis (GSEA) analysis by using gene-sets pathways from the KEGG and GO.

3. Wet lab experiments must be carried out to explore these hub genes to make the results of this study more convincing (e.g. PCR test of HaCaT cell line).

4. The author mentioned that the TOP2A was strongly associated with the survival of psoriasis patients. However, in the results part, I didn’t find any results over there. If so, I suggested the author could use the Kaplan–Meier estimator to make these results more convincing.

5. There are some minor language errors. The authors should be revised the manuscript with an English language editor to make it more readable.

Author Response

Thank you for your perspective comments on our paper, which are most helpful and valuable for improving the manuscript. After reconsidering seriously, our replies to the comments are as following. In addition, major revisions are indicated in red letters in the revised manuscript. 

Q1:

In the analysis of the differentially expressed genes (DEGs) analysis, the author used a threshold as |log2FC| ≥ 1.0, I suggest the author use a higher cutoff like 2 or 2.5 to show a higher uniqueness of these DEGs genes.

Our responses: 

As reviewer suggested in the differentially expressed genes (DEGs) analysis, we tried to use |log2FC| ≥ 2.0 or 2.5 to show the higher uniqueness of these DEGs genes before, but the Venn intersection did not filter out the ideal results. Therefore, we adopted |log2FC| ≥ 1.0 as the threshold for screening.

Q2:

In the functional enrichment analysis, the author only used an online website called DAVID to do the analysis. I suggest the author could also a try gene set enrichment analysis (GSEA) analysis by using gene-sets pathways from the KEGG and GO.

Our responses: 

As the reviewer’s suggestion, We have modified "Materials and Methods-4.3”. Gene ontology and pathway enrichment analysis according to request.

Q3:

Wet lab experiments must be carried out to explore these hub genes to make the results of this study more convincing (e.g. PCR test of HaCaT cell line).

Our responses: 

Thank you for you constructive suggestion, we did consider carry on a serious of experiments to explore these hub genes for making our results of this study more convincing. Unfortunatelly, with the COVID-19 epidemic raging around the world and China's epidemic prevention requirements, our laboratory in Yanbian University has not been opened yet. In this study, we screened the 10 central genes as TOP2A, CDKN3, MCM10, PBK, Gee, CEP55, ASPM, KIAA0101, ESC02, and IL-1. However, in our previous studies, we demonstrated that dihydroartemisinin (DHA)  inhibits IL-1β expression in psoriasis-like skin lesions. This consistents with the results of our screen. As soon as our laboratory returns to be normal state, we are looking forward to carry out some experiments with various methods (such as PCR, etc) to verify those hub genes. 

Q4:

The author mentioned that the TOP2A was strongly associated with the survival of psoriasis patients. However, in the results part, I didn’t find any results over there. If so, I suggested the author could use the Kaplan–Meier estimator to make these results more convincing.

Our responses: 

After cautiousexamination of the manuscript, we found we made a mistake in our presentation. TOP2A is a gene related to the cell cycle, it cannot be used as an indicator of survival in patients with psoriasis. Major revisions are indicated in red letters in the revised manuscript.

Abstract, v. 27-: (TOP2A) is a cell cycle-related gene that may have an impact on the progression of psoriasis by regulating the cell cycle.

Q5:

There are some minor language errors. The authors should be revised the manuscript with an English language editor to make it more readable.

Our responses: 

Thank you very much for the comment. As well as the reviewer indicated, we meticulously recheck all typos and grammar mistakes in the text and marked the corrected text.

Round 2

Reviewer 2 Report

The author did a really great answer to other questions, but I still insist that the wet lab experiments must be carried out.
